# Mediation Effect of Corporate Tax Burden and the Relationship between Environmental Regulation and Firm Performance

**DOI:** 10.3390/ijerph192214987

**Published:** 2022-11-14

**Authors:** Qiwen Dai, Huihua Huang, Xiaoqi Zhang, Yumin Su, Cheyuan Liu, Qiangyi Li

**Affiliations:** 1School of Economics and Management, Guangxi Normal University, Guilin 541004, China; 2The Qinzhou Central Sub-branch of the People's Bank of China, Qinzhou 535099, China; 3Institute of Economics, Chinese Academy of Social Science, Beijing 100732, China

**Keywords:** environmental regulation, firm performance, corporate tax burden, mediating effect, heavily polluting firms

## Abstract

This paper took the panel data of 1052 heavily-polluting listed companies from both the Shanghai and Shenzhen Stock Exchange from 2010 to 2017 to empirically analyze the impact of environmental regulation (ERG) on firm performance (FP). The article introduces a mediating effect model to test the mediating role of corporate tax burden (ETR) within the relationship between ERG on FP. The results showed that: (1) ERG has exerted a significant enhancement effect on the performance of heavily polluted firms via the ETR reduction mechanism. (2) The mediating effect of ETR depends on the duration of ERG. A significant time lag exists before the mediating effect starts to work, and the magnitude of the mediating effect increases with the time lag from the execution of the ERG. (3) The mediating effect of ETR varies significantly with the nature of corporate property rights. It is significant for the state-owned firms, while for non-state-owned firms, there is no evidence supporting the existence of the mediating effect of ETR despite ERG still having a significant direct-impact on FP. Based on these findings, we discuss the policy suggestion to optimize the impact of environmental regulation policies in terms of incentivizing the green development of polluting firms.

## 1. Introduction

Since the establishment of the reform and opening-up policy, China experienced the rapid growth of industrialization and urbanization driven by high investment and high energy consumption [1]. Despite the great progress in economic development, it also caused many problems, such as serious ecological degradation and environmental pollution. The Chinese central government has resorted to a variety of strategies to fight against environmental pollution and carbon emission, so as to coordinate economic growth and environmental improvement. Environmental regulation (ERG) is thought to be important and effective [2]. As a response to the growing voice for more stringent ERG, the Chinese government decided to implement more environmental protection policies. In 2017, at the 19th National Congress of the Communist Party of China (NPC), the need to cultivate a new growth engine and support high-quality development has been clearly stated. With the increasing attention on ecological protection, the relationship between ERG and firm performance (FP) is at the center of academic and policy debates. Although many empirical works have investigated the controversy, economists have not reached a consensus on the nature of the effect of ERG on FP or the mechanism that drives it.

In the early 1990s, Porter [3] and his collaborator challenged the conventional wisdom that ERGs increase firms’ environmental compliance costs, hence limiting their investment in other activities, ultimately inhibiting performance [3,4]. They provided a contrary view, also known as the Porter Hypothesis (PH), that more severe but well-designed ERGs can stimulate firms to innovate. Through cost management [5,6], the competitive advantages [7,8] and the pollution-related rent [9], firms can be fully or partially compensated for the cost incurred from the compliance with ERG, which helps improve performance. However, many empirical results show the opposite and imply that the PH does not hold. Some economists posited that ERG has no significant influence on FP [8,10,11]. Furthermore, some researchers concluded that ERG worsens corporate revenue and performance through an increase in the firm’s environmental costs [12], regulatory costs [13], and bargaining power [14]. One explanation for the counter-evidence to the PH is the duration of ERGs. ERG may negatively influence FP through increasing their environmental management costs in the short-term [15,16], and it can increase FP through improving the competitiveness effects in the long-term [17,18]. The negative effect of ERGs on FP also depends on the type of environmental management tools, i.e., ERGs had different impacts on firms’ financial distress [12]. In contrast to regulatory pressure, flexible regulation tools can optimize market structures, reduce production costs, and improve competitiveness, thus enhancing FP [8,14,19] that is independent from the time horizon [18].

Based on the brief literature review, a consensus in the academy has not yet been reached regarding the relation between ERGs and FP. Therefore, more comprehensive investigations are needed. Existing studies on this topic often begin from the perspectives of innovation capabilities [15,18,20,21], cost management and compliance costs [6], quality infrastructure [18], financial constraints, long-term investment behaviors, and capital–labor structures [16]. Few of them attempt to study the ERGs and FP from the perspective of ETR, which is crucial to the “win–win” situation for environmental protection and firm competitiveness. As an important factor affecting FP, corporate tax is one of the widely used instruments by which the government can implement ERG, which has strong incentives for a firm’s behavior. However, there is limited literature aiming at analyzing the impact of ERG via ETR. To fill this gap, in this paper, we explore the impacts of ERGs on FP, assess the mediating effects of ETR and try to provide more empirical evidence for the reference of policymakers.

Although heavy-polluting companies are the main targets of ERGs and the main practitioners of green governance, most of the literature on ERGs is at the industrial level, paying little attention to performance at the firm level. This study selects heavy-polluting listed companies in China from 2010 to 2017 as the sample and tests whether ERG promotes the performance of heavily polluting firms. We also study the channels by which ERG can affect FP in China. We concentrate on ETR and test whether it forms the causal link between ERG and FP.

This study contributes to the literature as follows: (1) We integrate ERG, ETR, and FP into a unified framework and analyze the impact of ERG on FP and the underlying mechanism, which enriches the literature; (2) Unlike the previous studies focusing on the industrial level, the analysis of this study is carried out on the firm level, which facilitates the assessment on firms’ behaviors against ERG and makes it possible to identify the micro-mechanism by which the ERG takes effect; (3) It is found that ETR, firm ownership, and the duration of ERG are the key variables shaping the relationship between ERG and FP.

The rest of this paper proceeds as follows: Section 2 presents the theoretical framework and hypotheses; Section 3 explains the data and methodology; Section 4 presents the empirical results, heterogeneity analyses, and robustness tests; and Section 5 includes the conclusions and discussions.

## 2. Theoretical Framework and Hypotheses

### 2.1. ERG and FP

Despite the growth literature on the effect of environmental policy on firm innovation and its relationship with FP, the academic community has not yet reached any consensus. The Porter hypothesis breaks through the traditional paradigm by focusing on the negative effects of ERG, and proposes a positive and optimistic view on the relationship between pollution control and its economic consequence [3,6]. This hypothesis has received extensive attention, and no consistent conclusion has been reached in literature. Evidence has been reported against the hypothesis and indicates that ERG cannot have a significant influence on competitiveness and financial performance [22,23,24]. Ambec and Barla [25], and Zheng and He [26] point towards it having a negative impact on productivity growth and the financial performance of pollution-intensive industries and firms. The imposition of China’s environmental protection tax has forced companies to increase their R&D investment, which inhibits the growth of FP in the short run [27]. The main reason for FP decline is that the relatively large elasticity of market demand prevents firms from passing regulatory costs to consumers [13]. Some scholars even questioned the fact that Porter’s hypothesis is only supported by a small number of company cases—it can hardly be generalized to the entire population of firms [11]. As a compromise, Jaffe and Palmer [28] presented three distinct variants of the Porter hypothesis: the “weak” version of the Porter hypothesis, the narrow Porter hypothesis, and the strong Porter hypothesis. Although the strong Porter hypothesis is difficult support, the results in Lanoie et al. [11] support the weak Porter hypothesis and the narrow Porter hypothesis. On the other hand, there is empirical evidence that supports and even reinforces the Porter hypothesis. Johnstone and Hallberg [29] and Agyemang et al. [30] found that the implementation of an environmental management system leads to improved financial and environmental outcomes, and the implementation of a national environmental policy has a significant incentive effect on the operating efficiency of Chinese small and medium-sized firms [31]. Reasonable ERG can promote competitiveness and performance through promoting firm innovation, reducing production costs, improving the internal agency problems, and improving resource utilization [6,7,32,33]. The “heterogeneity in compliance costs” of different sizes of enterprise can explain the impact of ERG on the company’s profit margins [34]. Zhang et al. [35] argued that there is an inverted U-shaped relationship between China’s environment regulation policy and the business performance of manufacturing firms. Different types of environmental management tools had different impacts on FP: Chen et al. [36] found that the command-control ERG suppresses FP, while the market ERG improves it.

Based on the brief review of the literature, we propose the following hypothesis:

**Hypothesis** **1.***ERG has a positive impact on FP*.

### 2.2. ERG and ETR

The tax distribution system is widely practiced in all major economies around the world. Its original intention was mainly to grant local governments the financial power to stimulate them to achieve economic growth goals [37]. The tax system has offered local governments considerable discretion over levying local taxes. This enables them to protect local firms by preferential tax policies and deduction policies [38,39], which provides flexibility in the actual tax burden for local firms. Stringent ERG will change a firm’s behaviors. Firms have to pay increased costs to meet the raised environmental standards. It demonstrated that firms in relatively disadvantageous positions have stronger incentives to avoid corporate tax due to high tax burden [40]. Tax avoidance is an economic behavior which may reduce costs directly. For example, Yu et al. [41] presented evidence to show that polluting firms may have more incentives to conduct tax avoidance activities, so as to help them retain economic resources, which is in line with their short-term interest [42]. On the other hand, when the stringent ERG from the central government or the external policies have negative impacts on the firms and regional economic growth, local governments may allow the local polluting firms to benefit from taxes and fees by implicitly relaxing the levy to supplement firms’ profits to deal with the pressure of ERG [41]. Under the stringent ERG, it is likely that local governments will seek more covert but legal protection for firms by tax discretion, such as tax avoidance, to reconcile the conflict between stringent ERG pressure and regional economic growth [41]. However, Ye and Lin [43] found that tough environmental regulation increases the tax rate, and this effect is positively related to policy strength and proportion of the secondary industry. The relationship between ERG and ETR is controversial.

Therefore, this paper proposes the following hypothesis:

**Hypothesis** **2.***ERG has a negative impact on ETR*.

### 2.3. ETR and FP

Taxation can be used to redistribute income, alter business structure, and maintain economic and resource balance [44]. While tax consequences are a motivating factor in many corporate decisions, the impact of business taxes and tax burdens on performance is controversial. Although some provinces have actually eliminated corporate taxes on small businesses or reduced such taxes to a symbolic level, there is no empirical evidence in favor of the effectiveness of such policies; small business growth is hampered by the existing tax system [45]. Sadeh et al. [46] also demonstrate that ETR stifle innovation and technological progress in OECD countries. When tax processes are simple and effectively organized and do not impose much burden on profits, firms have more incentives to invest, expand, and create jobs [47]. There are similar points of view in the literature. A tax structure that does not place a major cost on taxpayers could spur the movement of labor from energy-intensive to more energy-efficient industries, enhancing factor productivity and a society’s employment rate [48]. According to Nicolas and Loris [49], a business-friendly regulatory environment and tax system can reduce production cost and increase productivity. However, Saragih et al. [50] argued that the effective tax rate had a significant negative impact on a company’s growth prospects. Earnings performance did not weaken the negative effect of the effective tax rate on a company’s future growth opportunities. Park and Byun [51] found that corporate size and corporate group membership affect the relationship between the tax burden rate and future performance. The tax burden rate of small- and medium-sized enterprises had a negative effect on future performance, but not for large enterprises. Moreover, the tax burden ratio of companies not belonging to the corporate group has a significant negative relationship with future performance, but not for companies belonging to the corporate group.

Therefore, the following hypothesis can be deduced:

**Hypothesis** **3.***ETR and FP are negatively related*.

### 2.4. ERG, ETR, and FP

ERG can be implemented in a variety of different manners: there are direct/traditional regulation instruments (command-and-control), such as technology standards and non-tradable emission quotas, to indirect regulations (incentive-based/market-based), such as environmental taxes, green taxes, and tradable emission quotas [52]. Although direct regulation still predominates, economists have increasingly recognized the advantages of market-based ERG [53]. Empirical evidence from these studies suggests that indirect policy instruments will result in more cost-efficient emission reductions [54,55]. Both theoretical and empirical studies conclude that in contrast to direct regulation, indirect regulations will provide continuous dynamic incentives for firms to adopt technological improvement to reduce emissions [55,56]. Under the indirect ERG, ETR is an intermediate tool for the implementation of environmental policy, which causes FP to be affected by both regulation policy and its incurred tax burden. 

In order to verify the intermediary effect of ETR, based on H1-H3, hypothesis H4 is proposed:

**Hypothesis** **4.***ERG indirectly affect FP through the intermediary of ETR*.

The above analysis and the four hypotheses proposed can be summarized as the framework sketched in Figure 1, from which the key contribution of this paper is to verify the intermediary role of ETR that bridges the positive impact of ERG on FP. 

## 3. Data and Methodology

### 3.1. Data and Sample

We selected the listed heavy-polluting firms on the Shanghai and Shenzhen Stock Exchange from 2010 to 2017 as the preliminary samples. The selection of heavy-polluting firms was based on the research by Du et al. [57]. According to the “announcement on the implementation of special emission limits for atmospheric pollutants” issued by Ministry of Environmental Protection of People’s Republic of China, “Guidelines for Environmental Information Disclosure of Listed Companies” issued by Ministry of Environmental Protection of People’s Republic of China (2010)” and the “listed companies Industry Classification guidelines” issued by the Securities Regulatory Commission (2012), we selected firms from 11 industries to serve as the sample of heavy-polluting firms, including the petroleum and natural gas mining industry (B07), ferrous metal mining and separation industry (B08), non-ferrous metal mining and separation industry (B09), petroleum processing, coking and nuclear fuel processing industry (C25), chemical raw materials and chemical products manufacturing industry (C26), chemical fiber manufacturing industry (C28), rubber and plastic products industry (C29), non-metallic mineral products industry(C30), ferrous metal smelting and calendering industry (C31), nonferrous metal smelting and calendering industry (C32), and electricity, thermal production, and supply industry (D44). Furthermore, we filtered the samples according to the following criteria: (1) ST and SST type listed firms with abnormal financial status were exlcuded; (2) According to Wu [58], the pre-tax accounting profit being negative indicated that ETR is negative; it cannot reflect the real relationship between performance and the actual tax burden. Thus, the listed companies with negative pre-tax accounting profits during the study period were excluded; (3) Companies whose actual tax rate was less than 1 or greater than 0 during the study period were excluded as outliers. In addition, all continuous variables were Winsorized at 1% and 99%. Finally, according to the above criteria, 6762 observed values were obtained from 1052 heavy-polluting industrial firms. The data sources for the main variables in this paper included the following: the firm-level data were collected from the China Stock Market and Accounting Research (CSMAR) database and Wind database, and the measurement data on regional environment were taken from the China Statistical Yearbook and the China Environment Yearbook. 

### 3.2. Variable Measurements

#### 3.2.1. Dependent Variable: FP

We chose firms’ financial performance as the dependent variable in this study. According to Hutchinson and Gul [59], accounting-based measures are easily traced from the management ability to the firm value; they are best used in corporate governance empirical studies. There are similar points of view in the literature. The management aptitude towards asset efficiency and shareholders’ value can be reflected by higher FP [60]. Moreover, FP reveals firm production with relation to management while using assets. Therefore, by following the previous studies conducted by Rouf [61], Javeed et al. [62], and King and Lenox [63], we used FP measure as our dependent variable in this study, calculating it with net profit to total assets of the firm.

#### 3.2.2. Independent Variables: ERG

Calculating the ERG was critical because an accurate method of measurement significantly affects empirical studies [64]. Generally, ERG is considered as multidimensional, and any single mistake during measurement causes the wrong effects [62]. Therefore, in line with previous literature [23,65,66,67], we used the regional-level emissions of industrial wastewater, industrial sulfur dioxide, and industrial smoke and dust to calculate a comprehensive pollution emission index for the strictness degree of ERG in different regions. In detail, we take $Pollution_{ij}$ as the *j*-th type of pollution emission quantity (*j* = 1,2,3) in region *i* and calculate the regional pollution emission share $e_{ij}=n\times Pollution_{ij}/\overset{n}{\underset{i=1}{\sum }}Pollution_{ij}$. For every region *i*, we take the average $E_{i}=\left(e_{i1}+e_{i2}+e_{i3}\right)/3$ as the compound pollution share and take its inverse $ERG_{i}=1/E_{i}$ as the index measuring the strictness of ERG. As the stricter regulation implemented in region *i* often associates with a relatively lower pollutant emission share which leads to a greater $ERG_{i}$, therefore, $ERG_{i}$ is a well-defined measure for our analytic purpose.

#### 3.2.3. Mediating Variables: ETR

The effective tax rate (ETR) is defined as the ratio (in percentage) of taxes paid based on a firm’s current or total income to its pre-tax accounting income. Since ETR is used to measure a firm’s actual tax burden and tax holidays [68,69], this is a comprehensive reflection of the behaviors of firms and government in the process of tax collection and management. The definition of ETR has been widely varied in academic research [70]. There are two types of ETR in the theoretical studies: average ETR and marginal ETR [71]. Average ETR represents the overall tax burden on firms, and marginal ETR is used to investigate effects of taxation on a specific investment project. For the purpose of this article, we focus on average ETR. There are several ways to measure ETR [69,70,72,73,74], but no acceptable method has yet emerged [72]. The key issues here include how to determine tax expenses and how to calculate taxable income and the variability of ETR [75]. We follow the approach used by Porcano [69] and Liu and Cao [72] and define ETR as follows: ETR = (tax expense-deferred tax provision)/earnings before tax and interest. The smaller the ETR, the lower the corporate effective tax rate.

#### 3.2.4. Control Variables

Learning from the findings of Du et al. [57], Khan [76], Graham et al. [77], and Javeed et al. [62], this article has controlled for the influences of the following factors: the revenue growth rate (GROWTH), debt to assets ratio (LEVE), cash flow (CASHF), R&D investment intensity (RD), the ratio of book value versus market value (BM), firm size (SIZE), equity concentration (FIRST), the proportion of independent directors in the board (IND), the dummy for CEO duality (DUAL), and the dummy for the ownership (SOE) as control variables. Since SIZE and IND are highly collinear with other variables, they are eliminated from the regression model. The detailed definitions of the main variables in our empirical research are listed in Table 1.

### 3.3. Methods

Mediation analysis is important and frequently applied in studies in psychology and other social and behavioral science disciplines [78]. Different methods have been developed in testing mediation effects and in constructing their confidence intervals [79,80,81]. According to Wen and Fan [78] and MacKinnon et al. [82,83], for the sake of simplicity, we may only consider the standardized form of regression for the mediation models first. Assuming *X* is a predictor, *M* is a mediator, and *Y* is the dependent variable, typical mediation of *X* on *Y* via *M* can be modeled by the following equations: $Y=aX+e_{1}$, $M=bX+e_{2}$, $Y=cX+dM+e_{3}$. This indirect causal relationship (*X*→*M*→*Y*) is called the intermediary effect. Coefficient a measures the total effect of X on Y, b is the regression coefficient of *M* with respect to *X*, d is the regression coefficient of *Y* with respect to *M* after *X* is controlled, c is the regression coefficient of *Y* with respect to *X* after *M* is controlled, and *e*_1_ to *e*_3_ are the residuals of the regressions. c is the direct effect of *X* on *Y*, whereas the indirect effect b×d represents the mediation effect of *X* on *Y* through *M*. The total effect a should be equal to the mediating effect b×d plus the direct effect c. The proportion of intermediary effects is then measured by bda. To test the key hypotheses H1–H4 on the direct effect, indirect effect (mediation effect), and total effect of ERG on FP through income tax burden, we followed the approach used by Yan et al. [84], Tao et al. [85], and Wang et al. [86], and the following three regression equations were constructed, as follows: (1)$$ROA_{i,t}=\alpha_{0}+\alpha_{1}ERG_{i,t}+\alpha_{k}C{\mathrm{ontrol}}_{i,t}+\varepsilon_{i,t}$$
(2)$$ETR_{i,t}=\beta_{0}+\beta_{1}ERG_{i,t}+\beta_{k}C{\mathrm{ontrol}}_{i,t}+\varepsilon_{i,t}$$
(3)$$ROA_{i,t}=\delta_{0}+\delta_{1}ERG_{i,t}+\delta_{2}ETR_{i,t}+\delta_{k}C{\mathrm{ontrol}}_{i,t}+\varepsilon_{i,t}$$

In the formula, $ROA_{i,t}$ denotes the performance of firm i at year t; $ETR_{i,t}$ denotes the tax burden of firm i at year t; $ERG_{i,t}$ denotes the ERG of firm i at year t; *α_0_*, *β_0_*, and *δ_0_* are the intercept terms of the model; $\alpha_{1}$, $\beta_{1}$, $\delta_{1}$, $\delta_{2}$, $\alpha_{k}$, $\beta_{k}$, and $\delta_{k}$ are the constant to be estimated; $C{\mathrm{ontrol}}_{i,t}$ is the control variables associated with i and t, and $\varepsilon_{i,t}$ is the random disturbance term. The test for intermediary effect is implemented as in the following; Figure 2 shows the technical flowchart.

Firstly, test whether the total effect coefficient *α*_1_ of model (1) is significant. If it is significant, indicate that a mediating effect is confirmed; otherwise, it is a suppression effect. However, whether it is significant or not, follow-up testing is required. Secondly, test the significance of coefficient *β*_1_ in model (2) and the coefficient *δ*_2_ in model (3), which measure the mediating effect. If both are significant, the indirect effect is significant, and you can go directly to the fourth step. If at least one is not significant, proceed to step 3. Thirdly, test the significance of coefficient *β*_1_ × *δ*_2_ (H_0_: *β*_1_ × *δ*_2_ = 0) with the Bootstrap method [87]. If it is significant, the indirect effect is significant; otherwise, the indirect effect is not significant, and the test is stopped. Fourthly, test the significance of coefficient *δ*_1_ of the model (3), which measure the direct effect. If it is not significant, this indicates the existence of the complete mediating effect—that is, there is only a mediating effect and the direct effect is not significant. If it is significant, indicating that the direct effect is significant, proceed to the fifth step. Fifth, compare the signs of *β*_1_ × *δ*_2_, and *δ*_1_. If the signs are the same, this indicates a partial mediation effect is confirmed, and the proportion of the mediation effect (*β*_1_ × *δ*_2_/*α*_1_) needs to be reported. If the signs are different, the suppression effect is confirmed, and the absolute value of the ratio of indirect effect to direct effect (|*β*_1_ × *δ*_2_/*δ*_1_|) needs to be reported. We use the Sgmediation command in Stata14 software to test the mediation effect, and the significance test is made by Sobel test, Goodman test 1, Goodman test 2, and bootstrap test.

## 4. Empirical Results

### 4.1. Descriptive Statistics

Table 2 reports the descriptive statistics of variables. Q25% and Q75%, which represent the 25% quantile and the 75% quantile, respectively. The mean value of *FP* is 0.0801, and its median is 0.0661. These show that the economic performance of listed heavy-polluting firms generally follows a normal distribution, and the heterogeneity among different firms is obvious. Both the averaged strictness of *ERG* and the averaged tax burden are greater than their median, indicating that more firms are facing a mild *ERG* than the average and a relatively low *ETR*. Thus, *ERG* and *ETR* also present the above-mentioned normal distribution and heterogeneity characteristics. From the perspective of the ownership of firms, the averaged tax burden of the state-owned heavily polluting firms (0.3051) in the sample is much greater than that of non-state-owned heavily polluting firms (0.2392). Thus, we can draw the conclusion that the distribution of variables is within a reasonable range.

### 4.2. Correlation Analysis and Inter-Group Difference Test

The correlation analysis results are shown in Table 3, and the Spearman and Pearson correlation coefficients are listed in the upper right and lower left corners of the table, respectively. The Spearman/Pearson correlation coefficients between the *FP* and *ERG* indicator are 0.096 and 0.051, respectively, and the correlations between them are all significant at the 1% level. These show that without considering the influence of other factors, the more stringent the *ERG*, the easier it is to encourage heavily polluting firms to improve eco-performance, which essentially meets the expectations of Hypothesis 1. *FP* is significantly negatively correlated with *ETR* at the 1% level, the Spearman/Pearson correlation coefficients are −0.376 and −0.334, indicating that the lower *ETR* prompts up *FP*. The correlation coefficient between *ERG* and *ETR* is significantly negative, indicating that *ERG* has a significant impact on *ETR*, and the strengthening of *ERG* can reduce *ETR*. The correlation coefficients between other variables are also essentially in line with our expectations. In addition, we calculate the condition number statistics, which is 12.98 significantly less than the threshold 30. To prevent serious multicollinearity among explanatory variables, we also calculated the variance inflation factor (VIF value) of each variable. The calculation results show that the VIF value of each variable is less than 1.55, indicating that there is no serious multicollinearity among explanatory variables.

Concerning that the implementation of environmental policies takes time to affect *FP*, there exists a time lag before *ETR* starts to function. Therefore, variance analysis was used to examine the impact of *ERG* and their lags on the first, second, third, and fourth periods on *FP* and *ETR*. For each length of the lag, we divide firms into two groups according to the median of the regulation strictness they face, and compare the difference of the mean values in the performance and tax burden between the two groups of firms. Table 4 reports the test results of differences of performance and tax burden between corporate with different periods of *ERG*, which show that uniformly for each lag period, the mean value of *FP* is significantly higher for the group facing stricter *ERG*; while the mean value of *ETR* is significantly higher for the group facing less strict *ERG*. The test results are all significant at the 1% level. These observation supports the hypothesis of this paper to some extent that *ERG* has a positive effect on FP and a negative effect on *ETR* which won’t be affected by the lag period since the implementation of environmental policies. 

### 4.3. Empirical Results

#### 4.3.1. Main Analysis

Table 5 and Table 6 report the mediating effect of *ETR* on *ERG* and *FP*. In order to further validate the robustness of the result against the time lag of environmental policy, the first, second, third, and fourth lags of *ERG* are included in the equation. Model (1)–(3) in Table 5 associate with the regression coefficients in the three equations with the current ERG being utilized as the key explanatory variable, Model (4)–(6) in Table 5 and Model (7)–(15) in Table 6 report the result associated with time lags (L1–L4) of regulation policy. The regression results show that *ERG* have a significant impact on *FP* and *ETR*, but the direction of impact is significantly different. To test mediating effect of *ETR*, we need examine the direct effects of current and lagged *ERG* on *FP* firstly. In Models (1) and (4), the response of *FP* to *ERG* and the lagged *ERG* (i.e., the *L1. ERG*) are significantly positive at the confidential level of 5%, indicating that enhanced *ERG* will help improve *FP*. This test result supports the hypothesis H1. This finding agrees with Javeed et al. [62] and Johnstone and Hallberg [29], but differs from Xing et al. [10], López-Gamero et al. [8] and Lanoie et al. [11] that find that *ERG* has no significant influence on *FP*, and some researchers that concluded that *ERG* worsen corporate revenue and performance [12,13,14]. For the effect of *ERG* on *ETR*, we find in Model (2) and Model (5) that the response of *ETR* to *ERG* and *L1. ERG* is significant negative at the confidential level of 1% and 5%, respectively, indicating that the increasing strictness in *ERG* lower down *ETR*, which verifies H2. With testing the effect of *ETR* on *FP*, we find that the regression coefficients of *ETR* in Models (3) and (6) are both negative and significant at the confidential level of 1%, proving that *ETR* is an important factor affecting *FP*. This also validates hypothesis H3. These findings agree with McGuire et al. [88] and Adhikari et al. [89]. Compare with model (1), the relationship between *ERG* and *FP* in model (3) is not significant, indicating that *ERG* has no significant direct effect on *FP* in the current period, and *ETR* plays a full intermediary role between *ERG* and *FP*. Compared to Model (4), the regression coefficients and of *L1. ERG* in model (6) is still significant at the 10% confidential level, but the significance level and coefficient of *L1. ERG* reduce sharply, and the signs of *β*_1_
× *δ*_2_ and *δ*_1_ are the same, which reflect that the total impact of *ERG* is largely absorbed by the intermediary effect, and suggest that *ETR* plays a partial intermediary role between the lagged *ERG* and *FP*. This result supports the empirical analysis of Lee et al. [7], Xing et al. [33] and Zhou et al. [6] that *ERG* can help improve financial performance via mediating paths.

According to the intermediary effect test procedure [78,83], we also report the results of Sobel test, Goodman1 test and Goodman2 test. The mediating effect coefficients for both *ERG* and *L1. ERG* are much greater than the critical value (1.96) at the 5% confidential level, which validate the significance of the mediating effect of *ETR* and provide positive evidence to the hypothesis H4. As for the issues related to the relationship between *ERG* and *FP*, we integrate *ERG*, *ETR*, and *FP* into a unified framework and confirm that *ETR* is the underlying impact mechanism of *ERG* on *FP*, which enriches the literature in relation to the link between *ERG* and *FP*. Unlike the previous studies focusing on the industrial level, this paper analyses the relation between *ERG* and *FP* on the firm level, which facilitates the assessment on firms’ behaviors against to *ERG* and makes it possible to identify the micro-mechanism by which the *ERG* takes effect.

#### 4.3.2. The Impact of the Duration of ERG

The mediation effect of *ETR* varies with the length of time lag since the environmental policies were implemented. Same as Table 5, the regression results of model (7), (10), (13) in Table 6 indicate that *FP* and the *L2.ERG, L3.ERG* are significantly positive, while are insignificant in the fourth-lagged period. It shows that the total effect is significant in the second-lagged and the third-lagged period while has no obvious direct and indirect effects on *FP* in the fifth year of environmental policies. Model (8), (11), (14) in Table 6 indicate that the coefficients of the *L2.ERG, L3.ERG* are significantly negative, while are insignificant in the fourth-lagged period. It verifies H2 again. *ETR* and *FP* are significantly negative in model (9), (12), (15), which once again validates H3, and also indicating that *ERG* has significant indirect effect on *FP* in the second-lagged and the third-lagged period. The results of the Bootstrap test shows that the indirect effect of *ERG* in the fourth-lagged period is not significant (Table 7). Compared with model (7) and (10), the coefficients of *L2.ERG* and *L3.ERG* variables in model (9) and (12) are still significant, while the value and significance decrease markedly. Furthermore, the signs of *β*_1_
× *δ*_2_ and *δ*_1_ are the same. These show that *ETR* plays a partial mediating effect between *ERG* and *FP* in the second-lagged and the third-lagged period. These findings agree with Zhang and Du [16] that found that significant lagged effects of China’s environmental policies can be seen. It is revealed that the relationship between *ERG* and *FP* depends on the duration of *ERG*. The coefficient of *L4.ERG* in model (15) is not significant, which also confirms it has no significant impact on FP. The results of Sobel test, Goodman1 test and Goodman2 test are consistent with Table 5.

It can be seen from Table 5 and Table 6 that the mediating effects of *ETR*, measured by the coefficients and significance level of the Sobel test (mediating effect), is monotonically non-increasing with the time lag increasing from 0 (current) to 4. When the time lag is taken as 4, the Sobel Z-value is no longer significant coefficients even in the 10% confidential level. This fact shows that the current *ERG* is most conducive to exert the intermediary effect of *ETR*, while the role of *ERG* on the intermediary effect persists over a certain lag period. If we consider the proportion of intermediary effects in different periods, which are 34.23%, 20.89%, 18.21%, and 20.76% with respect to the regulation implemented in the current and lagged 1–3 years, the trend agrees with the variation in the significance level of Sobel test, suggesting the robustness of our result. The direct effect coefficients of each period of *ERG* are 0.00047, 0.00062, 0.00074, 0.00085, and 0.00064, respectively (Table 5 and Table 6), which are only significant for the first three lag period while are insignificant in the current and the fourth-lagged period. It is remarkable that both the scale and significance level of the direct effect coefficient reaches its maximum when the lag period is 3. This observation suggests that the direct impact of *ERG* on *FP* takes time to work which agrees with Hart et al. [17] and Peuckert [18] that find that it can increase *FP* through improving the competitiveness effects in the long-term and can be explained by the fact that firms need time to adapt to regulation policies, improve production processes, and re-organize productive resources. Finally, the total effect coefficients in the current and the lagged 1–3 period since the implementation of *ERG* are all significant at the confidential level of 5%, their estimated values are 0.00071, 0.00078, 0.00090, and 0.00107, respectively. This is consistent with the observation for both the direct and intermediary effect coefficients.

#### 4.3.3. The Impact of Firm Heterogeneity

Environmental policies impact on *FP* via the change on firm’s operational behaviors in response to the regulation. In practice, firm’s responsive behaviors are sensitive to the corporate culture, the management and innovation capabilities, and so on, which vary significantly with the ownership of firms [90]. In addition, the ownership of firms also significantly impacts *ETR*, which subsequently influences the intermediary effect of *ETR* [89]. Based on these concerns, we divide the full sample into two groups, state-owned firms and non-state-owned firms, to test whether the total effect of *ERG* and the intermediary effect of *ETR* are affected by the ownership structure (Table 8), where model (16)–(18) represents the three regression equations estimated within the state-owned subsample, model (19)–(21) represents the regression results for the non-state-owned subsample. In model (16), the regression coefficient of *ERG* is negative but not significant, indicating that the enhancement of *ERG* cannot effectively improve the performance of state-owned firms, which suggesting the existence of a “suppression effect” probably and need further inspection [91]. In Model (17), *ETR* negatively responses to *ERG* at the significance level of 1%, indicating that the actual tax burden of state-owned firms decreases as the intensity of *ERG* increases. In model (18), the regression coefficient of *ETR* is negative and are significant at the statistical level of 1%, proving that the lower tax burden for the state-owned firms leads to their better performance. According to the mediation effect test procedure [78,83], we find that the total effect coefficient in step 1 is not significant, while both of the coefficient $\beta_{1}$ and the coefficient $\delta_{2}$ in step 2 are significant and the *z* value of the mediating effect coefficient is greater than the 5% critical value in the Sobel test. However, the coefficient $\delta_{1}$ (*ERG*) in step 4 in equation (18) is not significant, which means that the direct effect is not significant, and there is only a mediating effect. These suggest that *ETR* plays a full intermediary role between *ERG* and *FP*. 

At the same time, the direct effect coefficient and total effect coefficient of *ERG* on the performance of state-owned firms are both negative but not significant, while the mediating effect coefficient of *ETR* is positive and significant at the 1% statistical level, the ratio of mediating effect is as high as 97.57%, which also confirms the above conclusion. This result provides support for that firm ownership is the key variables to shape the relationship between *ERG* and *FP*. It shows that there may be a negative mechanism between *ERG* and *FP* that has been concealed or there may be other intermediary paths [81,92]. The indirect effect of *ERG* affecting *SOE* performance through *ETR* weakens the direct impact, further lead to the total effect insignificant.

For the non-state-owned group, we find in model (19) that *FP* and *ERG* have a significant positive correlation at a statistical level of 1%, indicating that the stronger *ERG* leads to better performance of non-state-owned firms. This result supports the empirical analysis of Zhang et al. [31] that the promotion effect of *ERG* is easier to be seen in non-state-owned enterprises.

In model (20), the regression coefficients of *ERG* are positive but not significant, indicating that the enhancement of *ERG* cannot effectively reduce the tax burden of non-state-owned firms. In the model (21), the contribution of *ETR* to *FP* is negative and significant at the statistical level of 1%, proving that the reduction in the tax burden of non-state-owned firms brings the improvement of *FP*. Taken these findings together, we conclude that the intermediary effect of non-state-owned firms is not significant, the bootstrap test result in Table 7 also proves this point. On the other hand, the significant positivity of the direct effect and total effect coefficients can be witnessed, it shows that environmental regulation policies do contribute to the improvement of economic performance of non-state-owned firms, but this influence works straightforwardly, rather than via the pathway of corporate tax.

To this difference between state-owned and non-state-owned firms in response to *ERG*, we think it can be explained as the following: first, state-owned firms have various inherent advantages and the tax burden is heavier than that of non-state-owned firms. Under the constraints of *ERG*, they can help to reduce *ETR*. Secondly, due to the differences in positioning, development goals and personnel management of state-owned firms and non-state-owned firms [93,94], non-state-owned firms are not easy to pass *ETR* due to the constraints of scale, development level, management capabilities and innovation capabilities. The intermediary effect achieves the improvement of *ERG* goals and *FP*. At the same time, it may be easy for non-state-owned firms to pay attention to the constraints brought about by *ERG* and ignore the related preferential tax policies, making it difficult for *ETR* to play an intermediary role in the impact of *ERG* on *FP*. 

### 4.4. Robustness Testing

To guarantee the reliability of the analysis regarding the intermediary effect, Bootstrap test [87] are carried out to examine the robustness of above conclusions. It can be seen from Table 9 that the confidence intervals for the *ERG* group, *L1. ERG* group, *L2. ERG* group, *L3. ERG* group and the state-owned group do not contain 0 in their interior, indicating that there is an intermediary effect; while 0 is contained in the confidential interval for *L4. ERG* group and non-state-owned group, suggesting that the intermediary effect of *ETR* is not established, and the test result is consistent with the conclusion discussed in previous sections. 

At the same time, we also replace the dependent variable *FP* with *ROE* (Return on Equity) as the measure for *FP*. Except for the slight change in their absolute value, the sign and significance of the key parameter are consistent with the empirical results established so far (Table 10, Table 11 and Table 12), all hypotheses are supported, indicating that the conclusion is robust with respect to the measures of *FP*. Furthermore, the mediation effect coefficient shows a significant positive with the time lag increasing from 0 (current) to 4, and the direct effect coefficient and the total effect coefficient correspond with the findings. These show that the results are robust.

## 5. Conclusions and Discussion

### 5.1. Conclusions

With the increasing amount of attention given to environmental issues, the relationship between ERG and FP is a central question in environmental economics. Although many empirical works study this question, economists have not reached a consensus on the nature of the relationship or the mechanism that drives it. Therefore, this area motivates researchers to investigate and present an effective policy to control environmental issues. In this paper, we employ a mediation model to explore how ETR mediate the relationship between ERG and performance with a sample of 1052 listed heavy-polluting manufacturing firms for the period from 2010 to 2017. The findings are discussed and proposed as follows.

First, a close relationship exists among ERG, the tax burden of heavily polluting firms, and their economic performance. In the overall sense, ERG can effectively improve their performance. Strengthening the intensity of ERG will help reduce ETR, and reducing ETR will help improve the performance of firms. Due to the significant inhibitory effect of ETR on FP, the improved performance of heavily-polluted firms is mainly attributed to the reduction in the ETR induced by regulation-related preferential tax policies and subsidies. In other words, we find that ETR plays a full mediating role in the relationship between ERG and FP. After implementing environmental policies, tax relief will have a substantial impact on the R&D investment of firms in the long-term and short-term [95], which will help to promote the improvement in FP. In addition, local governments have effectively reduced the effective tax rate by means of illegal tax incentives, tax collection before repayment, and reduced tax law enforcement, in order to achieve the purpose of attracting the inflow of liquid production factors [96,97], which improve business performance. 

Second, the time when environmental regulation policies are implemented turns out influential to the total, direct and the intermediary effect, and there exists a positive time lag before these effects can reach their maximum. In the period immediately after its implementation, ERG does not have significant direct effect on FP, while its mediation effect through ETR is maximized in the sense that the total effect of ERG is completely taken up by the intermediary effect. The increase in the time lag makes the direct effect and total effect of ERG stronger and more significant; at the same time, its indirect impact on FP is still significantly present and keeps increasing in strength. The strength of all these direct and intermediary impacts, measured by the absolute value of the corresponding regression coefficients, reaches its maximum when the time lag is 3 years. Furthermore, the strengthening of ERG in the time lag reduces ETR, and the reduction in ETR improve FP. After the implementation of the environmental regulation policy, it may lead to an increase in costs for firms in the short term, for whom the failure to offset the compliance cost of ERG may not enable them to obtain benefits in the current period. At the same time, under the pressure of strict ERG, it takes a long time for firms to invest in R&D to improve business performance. All of these may lead to a lag in the direct impact of ERG on FP or the indirect impact through ETR.

Third, the implementation effect of ERG and the mediation effect of the tax burden are heterogeneous across firms, and vary significantly with the ownership of firms. From the perspective of the direct effect and total effect, enhancing ERG improves the performance of non-state-owned firms but has little impact on the performance of state-owned firms. State-owned firms have shown great advantages in financial support, resource allocation, financing capacity, and preferential policies [93], and they are not sensitive to the compliance cost brought about by environmental policies, since local governments have stronger incentives to provide them with financial subsidies. Due to the stiff industry competition of non-state-owned firms and the negative impact of environmental policies, the increased costs are quickly reflected in product prices [98]; they make up for the loss of legitimacy by improving FP and achieving a “win–win” of environmental and economic benefits. Moreover, as found by Wang et al. [99] and Wang and Wheeler [100], state-owned firms actually face weaker environmental regulatory constraints, i.e., it is easier for them to evade regulation, while non-state-owned firms lack bargaining power for ERG and have higher environmental violation costs. From the perspective of the intermediary effect, ERG can significantly reduce the tax burden of state-owned firms but does not affect the tax burden of non-state-owned firms. As a consequence, the mediation effect via the tax burden for state-owned firms is well-established, but not for non-state-owned firms. Firms may choose to mitigate the compliance cost increase through more tax avoidance activities, to retain economic resources and survive when they encounter stringent ERG [41]. Due to the flexibility of non-state-owned firms, they can reduce ETR through economic risk assessment, profit margin control, and tax planning to reduce their costs and improve competitiveness and performance. Therefore, ERG does not significantly promote the performance of non-state-owned heavily-polluting firms by reducing ETR. However, under government intervention, managers of local state-owned firms are often appointed and subject to assessment by local governments [94], which reduces the degree of information asymmetry between local governments and local state-owned firms. This makes it difficult for local state-owned firms to evade tax burdens by hiding their income. Due to the insignificant total effect, the intermediary effect for state-owned companies is suppressed by the direct effect of ERG. However, the reduction in the tax burden of both state-owned and non-state-owned firms can help improve their performance, which supports the empirical analysis of Adhikari et al. [89].

### 5.2. Practical Implications

Our study has several practical implications. Environmental regulation policies are the preconditions for environmental governance, and reasonable institutional arrangements and appropriate policy requirements are conducive to the implementation of environmental policies. According to Zhang et al.’s [101] findings, environmental problems are mainly led by high pollution- and high energy-consumption firms for pursuing huge profits. Thus, curbing the expansion of heavy-polluting firms and encouraging them to realize green transformation are powerful moves to alleviate pollutant emissions at the source. For policymakers, especially on the premise of not reducing the economic benefits of firms, it is more helpful for firms to actively implement pro-environmental behaviors. The design of environmental regulation policy needs to consider the feasibility of policy objectives and the tolerance level of firms in different periods. Increase ERG on high-polluting firms while increasing environmental subsidies to motivate firms that comply with pollution discharge standards and actively implement environmental protection behaviors. Comprehensively evaluate the effects of ERG, dynamically adjust and optimize environmental policies, and improve policy flexibility. Theoretically, the reduction in tax burden contingent on the adoption of environmentally-friendly production technology helps improve environmental quality while minimize the negative externality to firms. Use market-based environmental policy tools to provide firms with more space to find technical solutions to reduce compliance costs and guide and incentivize firms to take actions to reduce ETR or technological innovation and improve business performance, which is more conducive to firm innovation than command-and-control regulatory methods [102]. Strengthen policy support for non-state-owned heavy polluting firms and encourage them to implement environmental technology innovation and R&D investment. It is worth noting that the effects of environmental regulation policies implemented in different time periods are different, so specific policies need to be formulated according to each time point.

In order to promote the green development of firms, make firms comply with environmental regulation policies, and actively reduce pollution emissions, the government should issue a series of tax preferential policies to assist in the implementation of environmental regulation policies. For example, firms can enjoy the favorable tax rate as a subsidy to their investment in technology R&D and the costs of purchasing environmental protection and energy-saving equipment, which reduces ETR and offsets the impact of environmental costs on FP. Local governments need to actively lower their tax collection efforts to protect firms from too high costs. The tax distribution system in market economy countries may encourage the local governments to have a partiality to protect local firms [39] and also enable them to make tax preferences and deductions policies, even implicitly [38]. Successful ERG can not only achieve the effect of ecological protection and emission reduction but also improve the competitiveness of firms, which can improve resource utilization efficiency and increase a firm’s profits. FP not only encourages the firm towards green innovative practices but also plays positive role for positivity amid proactive environmental strategies and green innovation [103].

### 5.3. Limitations and Future Research

There are some limitations that could be addressed in future research. First, due to the relatively difficult acquisition of corporate data, the construction of relevant variables is relatively simple, and there are certain limitations. For example, measuring the economic performance of a firm by the return on assets may not be comprehensive enough, which lacks consideration of their growth potential. Subsequent research can construct more comprehensive performance indicators to push the analysis forward. Second, multiple types of policy tools exist for the implementation of ERG, such as mandatory ERG and voluntary ERG, which impact FP in quite different way. It is worth thinking about analyzing the impact and its pathways of different types of ERG on FP. Third, the ETR is measured only by corporate income tax in this article. However, beyond income tax, the government collects many other taxes, which can also guide a firm’s behavior and facilitate the implementation of environmental protection. These non-income taxes include the value-added tax, resource tax, environmental tax, and so on. How these taxes affect FP and react to the implementation of ERG is not yet addressed in this article, which calls for future studies. Finally, the “suppression effect” needs to be further considered, which implies that the negative impact of ERG is still not negligible and how to design ERG to achieve both the policy objectives and the green development of polluting firms is an open question that remains to be discussed.

## Figures and Tables

**Figure 1 ijerph-19-14987-f001:**
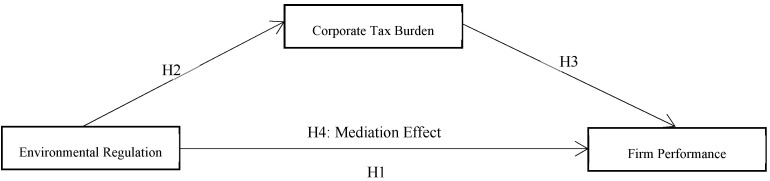
Hypothesized model.

**Figure 2 ijerph-19-14987-f002:**
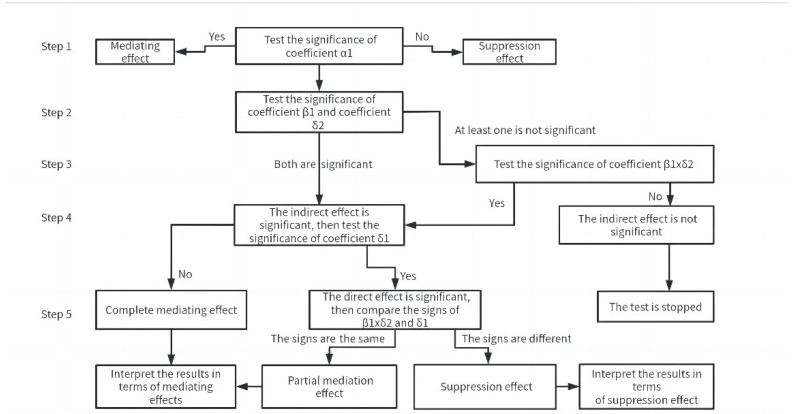
Technical flowchart.

**Table 1 ijerph-19-14987-t001:** Definition of variables.

Variable Category	Variable Names	Abbreviations	Measures	Expected Priori
Dependent Variable	Firm performance	*FP*	Net profits divided by total assets	+
Independent Variable	Environmental regulation	*ERG*	Intensity index based on three types of waste emissions data	
Mediating Variable	Corporate tax burden	*ETR*	Difference between tax expense and deferred tax provision divided by earnings before tax and interest	−
Control Variables	Operating income growth rate	*GROWTH*	The difference between the operating income of this year and the previous year divided by last year’s operating income	+
	Leverage	*LEVE*	Total liabilities divided by total assets at end of period	−
	Cash flow	*CASHF*	Net cash flows from operating activities divided by operating income	+
	R&D investment intensity	*RD*	Ratio of R&D investment to operating income	−
	Book to market ratio	*BM*	Total assets divided by market value	−
	Equity concentration	*FIRST*	Proportion of the largest shareholder’s stock holding	+
	At the same time as chairman and general manager	*DUAL*	The chairman and general manager are assumed to be one by the same person, otherwise zero	−
	Firm ownership	*SOE*	Taking the value of one if the firm if a state-owned firm, zero otherwise.	+

+ denotes a positive correlation and − denotes a negative correlation.

**Table 2 ijerph-19-14987-t002:** Descriptive statistical analysis.

Variable	Obs.	Mean			Min	Q_25%_	Med	Q_75%_	Max	S.D.
		Sample	State-Owned	Non-State						
*FP*	6762	0.0801	0.0569	0.0910	−0.1222	0.0325	0.0661	0.1124	0.3254	0.0662
*ERG*	6762	2.3964	2.4455	2.3733	0.4056	1.3844	2.0882	2.9595	10.0917	1.7328
*ETR*	6762	0.2603	0.3051	0.2392	0	0.1628	0.2144	0.3094	0.9989	0.1667
*GROWTH*	6762	0.1688	0.1436	0.1806	−0.4903	0.0029	0.1230	0.2609	1.7380	0.2760
*LEVE*	6762	0.3989	0.4818	0.3599	0.0075	0.2383	0.3885	0.5456	0.9146	0.2007
*CASHF*	6756	0.1214	0.1354	0.1148	−0.6844	0.0406	0.1062	0.1892	0.6902	0.1536
*RD*	6756	0.0248	0.0145	0.0297	0	0.0019	0.0231	0.0372	0.2173	0.0253
*BM*	5255	0.3825	0.4653	0.3299	0.0150	0.2061	0.3271	0.4956	1.2469	0.2432
*FIRST*	5546	0.3705	0.4047	0.3502	0.0880	0.2564	0.3578	0.4717	0.8000	0.1530
*DUAL*	6737	0.3083	0.1100	0.4000	0	0	0	1	1	0.4618
*SOE*	6762	0.3197	1	0	0	0	0	1	1	0.4664

**Table 3 ijerph-19-14987-t003:** Correlation coefficient matrix of variables.

Variable	*FP*	*ERG*	*ETR*	*GROWTH*	*LEVE*	*CASHF*	*RD*	*BM*	*FIRST*	*DUAL*	*SOE*
*FP*	1	0.096 ***	−0.376 ***	0.200 ***	−0.407 ***	0.307 ***	0.193 ***	−0.348 ***	0.108 ***	0.177 ***	−0.290 ***
*ERG*	0.051 ***	1	−0.087 ***	0.006	−0.121 ***	−0.012 ***	0.180 ***	0.002	0.002	0.122 ***	−0.114 ***
*ETR*	−0.334 ***	−0.056 ***	1	−0.143 ***	0.209 ***	−0.101 ***	−0.170 ***	0.161 ***	0.014	−0.084 ***	0.185 ***
*GROWTH*	0.177 ***	0.008	−0.138 ***	1	0.013	−0.066 ***	0.081 ***	−0.126 ***	−0.010	0.009	−0.068 ***
*LEVE*	−0.362 ***	−0.065 ***	0.207 ***	0.043 ***	1	−0.136 ***	−0.387 ***	0.196 ***	0.038 ***	−0.123 ***	0.279 ***
*CASHF*	−0.282 ***	0.040 ***	−0.089 ***	−0.047 ***	−0.096 ***	1	0.037 ***	−0.048 ***	0.060 ***	0.008	0.037 ***
*RD*	0.180 ***	0.097 ***	−0.142 ***	0.034 ***	−0.358 ***	0.068 ***	1	−0.169 ***	−0.083 ***	0.167 ***	−0.298 ***
*BM*	−0.326 ***	0.025 *	0.161 ***	−0.121 ***	0.211 ***	0.013	−0.218 ***	1	0.105 ***	−0.112 ***	0.249 ***
*FIRST*	0.112 ***	0.076 ***	0.004	−0.004	0.043 ***	0.097 ***	−0.092 ***	0.153 **	1	−0.011	0.165 ***
*DUAL*	0.158 ***	−0.089 ***	−0.089 ***	0.035 *	−0.126 ***	−0.010	0.176 ***	−0.124 ***	−0.027 **	1	−0.296 ***
*SOE*	−0.240 ***	0.184 ***	0.184 ***	−0.062 ***	0.283 ***	0.062 ***	−0.280 ***	0.271 ***	0.172 ***	−0.296 ***	1

Notes: The upper/lower triangle is the Spearman/Pearson correlation coefficient. *, **, and *** represent significance at the levels of 10%, 5%, and 1%, respectively.

**Table 4 ijerph-19-14987-t004:** Test of differences between corporate with different periods of ERG.

	*Mean FP*	*Mean ETR*		*Mean FP*	*Mean ETR*
*ERG* < median (*n* = 3337)	0.0764	0.2731	*L1. ERG* < median (*n* = 2650)	0.0788	0.2645
*ERG* > median (*n* = 3425)	0.0836	0.2477			
*t*-test	−0.0072 *** (−4.49)	0.0254 *** (6.27)	*L1. ERG* > median (*n* = 2679)	0.0835	0.2442
*L2. ERG* < median (*n* = 2148)	0.0742	0.2726	*t*-test	−0.0047 *** (−2.76)	0.0203 *** (4.93)
*L2. ERG* > median (*n* = 2225)	0.0799	0.2514	*L3. ERG* < median (*n* = 1691)	0.0715	0.2805
*t*-test	−0.0057 *** (−3.09)	0.0212 *** (4.39)			
*L4. ERG* < median (*n* = 1349)	0.0678	0.2844	*L3. ERG* > median (*n* = 1851)	0.0782	0.2554
*L4. ERG* > median (*n* = 1413)	0.0733	0.2628	*t*-test	−0.0067 *** (−3.43)	0.0252 *** (4.58)
*t*-test	−0.0055 ** (−2.57)	0.0215 *** (3.25)			

Notes: Robust t-statistics are in parentheses; *, **, and *** represent significance at the levels of 10%, 5%, and 1%, respectively.

**Table 5 ijerph-19-14987-t005:** Regression results (current period and lag phase I).

Variables	Dependent Variables: *ERG*			Dependent Variables: *L1. ERG*		
	*(1) FP*	*(2) ETR*	*(3) FP*	*(4) FP*	*(5) ETR*	*(6) FP*
*Constant*	0.0948 ***	0.2345 ***	0.1080 ***	0.0942 ***	0.2360 ***	0.1071 ***
	(34.82)	(23.14)	(38.65)	(32.66)	(23.39)	(35.62)
*ERG*	0.0007 **	−0.0043 ***	0.0005			
	(1.98)	(−3.24)	(1.33)			
*L. ERG*				0.0008 **	−0.0030 **	0.0006 *
				(2.14)	(−2.32)	(1.72)
*ETR*			−0.0566 ***			−0.0549 ***
			(−15.53)			(−12.72)
*GROWTH*	0.0393 ***	−0.0872 ***	0.0343 ***	0.0459 ***	−0.1071 ***	0.0400 ***
	(17.33)	(−10.39)	(15.53)	(18.07)	(−12.04)	(15.78)
*LEVE*	−0.0967 ***	0.1402 ***	−0.0887 ***	−0.0936 ***	0.1444 ***	−0.0857 ***
	(−27.78)	(10.88)	(−25.78)	(−24.69)	(10.87)	(−22.71)
*CASHF*	0.0814 ***	−0.0821 ***	0.0764 ***	0.0880 ***	^−0.1042 ***^	0.0823 ***
	(19.87)	(−5.41)	(19.11)	(19.99)	(−6.76)	(18.94)
*RD*	−0.0683 **	−0.2978 ***	−0.0852 ***	−0.0573 *	−0.3358 ***	−0.0757 **
	(−2.33)	(−2.74)	(−2.97)	(−1.88)	(−3.15)	(−2.53)
*BM*	−0.0581 ***	0.0619 ***	−0.0546 ***	−0.0602 ***	0.0423 ***	−0.0580 ***
	(−20.75)	(5.97)	(−19.88)	(−20.11)	(4.03)	(−19.66)
*FIRST*	0.0423 ***	−0.0166	0.0414 ***	0.0413 ***	−0.0107	0.0407 ***
	(9.62)	(−1.02)	(9.61)	(8.84)	(−0.65)	(8.88)
*DUAL*	−0.0025	−0.0010	−0.0025 *	−0.0033 **	0.0033	−0.0031 **
	(−1.62)	(−0.19)	(−1.69)	(−2.08)	(0.60)	(−2.01)
*SOE*	−0.0012	0.0254 ***	0.0002	0.0008	0.0187 ***	0.0018
	(−0.81)	(4.56)	(0.15)	(0.51)	(3.33)	(1.16)
*VIF*	<2	<2	<2	<2	<2	<2
*Adj-R²*	0.3305	0.0936	0.3599	0.3599	0.1105	0.3833
*F*	289.16 ***	61.28 ***	296.29 ***	265.46 ***	59.45 ***	264.20 ***
*Observations*	5254	5254	5254	4235	4235	4235
*Sobel*	0.00024 *** (*z* = 3.170)			0.00016 ** (*z* = 2.287)		
*Goodman* 1	0.00024 *** (*z* = 3.164)			0.00016 ** (*z* = 2.280)		
*Goodman 2*	0.00024 *** (*z* = 3.176)			0.00016 ** (*z* = 2.294)		
Mediating effect	0.00024 *** (*z* = 3.170)			0.00016 ** (*z* = 2.287)		
Direct effect	0.00047 (*z* = 1.333)			0.00062 * (*z* = 1.721)		
Total effect	0.00071 ** (*z* = 1.984)			0.00078 ** (*z* = 2.138)		
Ratio of mediating effect	0.3423			0.2089		

Notes: Robust *t*-statistics are in parentheses; *, **, and *** represent significance at the levels of 10%, 5%, and 1%, respectively.

**Table 6 ijerph-19-14987-t006:** Regression results (lag phase II, III and IV).

Variables	Dependent Variables: *L2. ERG*			Dependent Variables: *L3. ERG*			Dependent Variables: *L4. ERG*		
	*(7) FP*	*(8) ETR*	*(9) FP*	*(10) FP*	*(11) ETR*	*(12) FP*	*(13) FP*	*(14) ETR*	*(15) FP*
*Constant*	0.0918 ***	0.2285 ***	0.1040 ***	0.0898 ***	0.2308 ***	0.1043 ***	0.0836 ***	0.2400 ***	0.0977 ***
	(29.44)	(19.86)	(32.32)	(27.13)	(18.50)	(30.75)	(22.71)	(16.70)	(25.89)
*L2. ERG*	0.0009 **	−0.0030 **	0.0007 *						
	(2.34)	(−2.16)	(1.95)						
*L3. ERG*				0.0011 **	−0.0035 **	0.0008 **			
				(2.52)	(−2.21)	(2.05)			
*L4. ERG*							0.0007	−0.0023	0.0006
							(1.62)	(−1.24)	(1.37)
*ETR*			−0.0536 ***			−0.0628 ***			−0.0588 ***
			(−12.10)			(−13.47)			(−11.91)
Control variables	*Y*	*Y*	*Y*	*Y*	*Y*	*Y*	*Y*	*Y*	*Y*
*VIF*	<2	<2	<2	<2	<2	<2	<2	<2	<2
*Adj-R²*	0.3652	0.1122	0.3899	0.3649	0.1207	0.4004	0.3658	0.1151	0.3990
*F*	231.33 ***	51.59 ***	231.28 ***	196.49 ***	47.71 ***	205.45 ***	164.74 ***	37.93 ***	170.64 ***
*Observations*	3605	3605	3605	3063	3063	3063	2556	2556	2556
*Sobel*	0.00016 ** (*z* = 2.124)			0.00022 ** (*z* = 2.177)			0.00014 (*z* = 1.229)		
*Goodman* 1	0.00016 ** (*z* = 2.117)			0.00022 ** (*z* = 2.172)			0.00014 (*z* = 1.225)		
*Goodman 2*	0.00016 ** (*z* = 2.131)			0.00022 ** (*z* = 2.183)			0.00014 (*z* = 1.233)		
Mediating effect	0.00016 ** (*z* = 2.124)			0.00022 ** (*z* = 2.177)			0.00014 (*z* = 1.229)		
Direct effect	0.00074 * (*z* = 1.955)			0.00085 ** (*z* = 2.051)			0.00064 (*z* = 1.372)		
Total effect	0.00090 ** (*z* = 2.345)			0.00107 ** (*z* = 2.518)			0.00078 (*z* = 1.620)		
Ratio of mediating effect	0.1821			0.2076			0.1752		

Notes: Robust t-statistics are in parentheses; *, **, and *** represent significance at the levels of 10%, 5%, and 1%, respectively.

**Table 7 ijerph-19-14987-t007:** The bootstrap test results.

	*ERG*	*L1. ERG*	*L2. ERG*	*L3. ERG*	*L4. ERG*	State-Owned Firms	Non-State-Owned Firms
bs1	0.0003 ***	0.0002 **	0.0002 **	0.0003 **	0.0002	0.0004 ***	−0.0002
	(3.27)	(2.37)	(2.23)	(2.18)	(1.19)	(3.58)	(−1.03)
bs2	0.0005	0.0007 *	0.0008 **	0.0009 *	0.0006	−0.0007 *	0.0024 ***
	(1.41)	(1.70)	(2.04)	(1.81)	(1.27)	(−1.79)	(3.55)

Notes: Robust t-statistics are in parentheses; *, **, and *** represent significance at the levels of 10%, 5%, and 1%, respectively.

**Table 8 ijerph-19-14987-t008:** Considering Firm ownership.

Variables	Dependent Variables: *ERG* (State-Owned Firms)			Dependent Variables: *ERG* (Non-State-Owned Firms)		
	*(16) FP*	*(17) ETR*	*(18) FP*	*(19) FP*	*(20) ETR*	*(21)* *FP*
*Constant*	0.0923 ***	0.2996 ***	0.1064 ***	0.0960 ***	0.1831 ***	0.1086 ***
	(22.29)	(16.47)	(24.66)	(24.18)	(14.24)	(27.23)
*ERG*	−0.0002	−0.0082 ***	−0.0006	0.0019 ***	0.0027	0.0021 ***
	(−0.38)	(−4.17)	(−1.27)	(3.31)	(1.44)	(3.73)
*ETR*			−0.0471 ***			−0.0691 ***
			(−9.53)			(−12.72)
Control variables	*Y*	*Y*	*Y*	*Y*	*Y*	*Y*
*VIF*	<2	<2	<2	<2	<2	<2
*Adj-R²*	0.3312	0.0727	0.3596	0.3083	0.0891	0.3428
*F*	127.16 ***	20.97 ***	128.14 ***	180.07 ***	40.31 ***	187.29 ***
*Observations*	2039	2039	2039	3215	3215	3215
*Sobel*	0.0004 *** (*z* = 3.822)			−0.0002 (*z* = −1.433)		
*Goodman* 1	0.0004 *** (*z* = 3.804)			−0.0002 (*z* = −1.429)		
*Goodman 2*	0.0004 *** (*z* = 3.839)			−0.0002 (*z* = −1.438)		
Mediating effect	0.0004 *** (*z* = 3.822)			−0.0002 (*z* = −1.433)		
Direct effect	−0.0006 (*z* = −1.270)			0.0021 *** (*z* = 3.731)		
Total effect	−0.0002 (*z* = 0.701)			0.0019 *** (*z* = 3.315)		
Ratio of mediating effect	0.9757			0.0975		

Notes: Robust t-statistics are in parentheses; *, **, and *** represent significance at the levels of 10%, 5%, and 1%, respectively.

**Table 9 ijerph-19-14987-t009:** The bootstrap robustness tests.

Group	*β*	*Boot SE*	95% Confidence Interval	
			Lower Limit	Upper Limit
*ERG*	0.00024	0.00008	0.00008	0.00040
*L1. ERG*	0.00016	0.00007	0.00002	0.00032
*L2. ERG*	0.00016	0.00008	0.00001	0.00033
*L3. ERG*	0.00022	0.00010	0.00003	0.00043
*L4. ERG*	0.00013	0.00012	−0.00009	0.00018
State-owned firms	0.00039	0.00011	0.00020	0.00062
Non state-owned firms	−0.00019	0.00014	−0.00047	0.00009

Notes: Since the nonparametric percentile Bootstrap sampling self-help method for deviation correction more accurately reflects the regression results, the confidence intervals listed in the table above are deviation correction confidence intervals.

**Table 10 ijerph-19-14987-t010:** Robustness testing of different measurement indicators (current period and lag phase I).

Variables	Dependent Variables: *ERG*			Dependent Variables: *L1. ERG*		
	*(1)* *ROE*	*(2)* *ETR*	*(3)* *ROE*	*(4)* *ROE*	*(5)* *ETR*	*(6)* *ROE*
*Constant*	0.0980 ***	0.2345 ***	0.1196 ***	0.0988 ***	0.2360 ***	0.1237 ***
	(20.04)	(23.14)	(23.71)	(19.88)	(23.39)	(23.98)
*ERG*	0.0011	−0.0043 ***	0.0006			
	(1.63)	(−3.24)	(1.03)			
*L1. ERG*				0.0010	−0.0030 **	0.0007
				(1.64)	(−2.32)	(1.17)
*ETR*			−0.0922 ***			−0.1057 ***
			(−14.02)			(−14.29)
Control variables	*Y*	*Y*	*Y*	*Y*	*Y*	*Y*
*VIF*	<2	<2	<2	<2	<2	<2
*Adj-R²*	0.2004	0.0936	0.2292	0.2284	0.1105	0.2638
*F*	147.32 ***	61.28 ***	157.18 ***	140.25 ***	59.45 ***	152.69 ***
*Observations*	5254	5254	5254	4235	4235	4235
*Sobel*	0.00040 *** (*z* = 3.155)			0.00031 ** (*z* = 2.294)		
*Goodman* 1	0.00040 *** (*z* = 3.147)			0.00031 ** (*z* = 2.289)		
*Goodman 2*	0.00040 *** (*z* = 3.163)			0.00031 ** (*z* = 2.300)		
Mediating effect	0.00040 *** (*z* = 3.155)			0.00031 ** (*z* = 2.287)		
Direct effect	0.00066 (*z* = 1.033)			0.00072 (*z* = 1.721)		
Total effect	0.00105 (*z*= 1.631)			0.00103 (*z* = 1.644)		
Ratio of mediating effect	0.3773			0.2089		

Notes: Robust t-statistics are in parentheses; *, **, and *** represent significance at the levels of 10%, 5%, and 1%, respectively.

**Table 11 ijerph-19-14987-t011:** Robustness testing of different measurement indicators (lag phase II, III and IV).

Variables	Dependent Variables: *L2. ERG*			Dependent Variables: *L3. ERG*			Dependent Variables: *L4. ERG*		
	*(7)* *ROE*	*(8)* *ETR*	*(9)* *ROE*	*(10)* *ROE*	*(11)* *ETR*	*(12)* *ROE*	*(13)* *ROE*	*(14)* *ETR*	*(15)* *ROE*
*Constant*	0.0958 ***	0.2285 ***	0.1155 ***	0.0913 ***	0.2308 ***	0.1189 ***	0.0821 ***	0.2400 ***	0.1080 ***
	(17.91)	(19.86)	(20.83)	(15.83)	(18.50)	(20.25)	(12.37)	(16.70)	(15.89)
*L2. ERG*	0.0015 **	−0.0031 **	0.0013 **						
	(2.34)	(−2.16)	(1.98)						
*L3. ERG*				0.0019 ***	−0.0035 **	0.0015 **			
				(2.62)	(−2.21)	(2.12)			
*L4. ERG*							0.0015 *	−0.0023	0.0012
							(1.76)	(−1.24)	(1.51)
*ETR*			−0.0858 ***			−0.1198 ***			−0.1080 ***
			(−11.25)			(−14.83)			(−12.14)
Control variables	*Y*	*Y*	*Y*	*Y*	*Y*	*Y*	*Y*	*Y*	*Y*
*VIF*	<2	<2	<2	<2	<2	<2	<2	<2	<2
*Adj-R²*	0.2303	0.1122	0.2563	0.2175	0.1207	0.2699	0.2129	0.1151	0.2557
*F*	120.83 ***	51.59 ***	125.20 ***	95.58 ***	47.71 ***	114.17 ***	77.78 ***	37.93 ***	88.77 ***
*Observations*	3605	3605	3605	3063	3063	3063	2556	2556	2556
*Sobel*	0.00026 ** (*z* = 2.119)			0.00042 ** (*z* = 2.182)			0.00025 (*z* = 1.229)		
*Goodman* 1	0.00026 ** (*z* = 2.111)			0.00042 ** (*z* = 2.178)			0.00025 (*z* = 1.225)		
*Goodman 2*	0.00026 ** (*z* = 2.127)			0.00042 ** (*z* = 2.187)			0.00025 (*z* = 1.234)		
Mediating effect	0.00026 ** (*z* = 2.119)			0.00042 ** (*z* = 2.182)			0.00025 (*z* = 1.230)		
Direct effect	0.00128 ** (*z* = 1.975)			0.00152 ** (*z* = 2.123)			0.00127 (*z* = 1.509)		
Total effect	0.00154 ** (*z* = 2.341)			0.00194 ** (*z* = 2.625)			0.00152 (*z* = 1.757)		
Ratio of mediating effect	0.1700			0.2179			0.1645		

Notes: Robust t-statistics are in parentheses; *, **, and *** represent significance at the levels of 10%, 5%, and 1%, respectively.

**Table 12 ijerph-19-14987-t012:** Robustness testing of different measurement indicators (Considering Firm ownership).

Variables	Dependent Variables: *ERG* (State-Owned Business)			Dependent Variables: *ERG* (Non-State-Owned Firms)		
	*(16) ROE*	*(17) ETR*	*(18) ROE*	*(19) ROE*	*(20) ETR*	*(21) ROE*
*Constant*	0.0882 ***	0.2996 ***	0.1139 ***	0.1031 ***	0.1831 ***	0.1211 ***
	(11.05)	(16.47)	(13.68)	(15.03)	(14.24)	(17.41)
*ERG*	−0.0002	−0.0082 ***	−0.0007	0.0023 **	0027	0.0026 **
	(−0.02)	(−4.17)	(−0.85)	(2.27)	(1.44)	(2.58)
*ETR*			−0.0860 ***			−0.0982 ***
			(−9.01)			(−10.61)
Control variables	*Y*	*Y*	*Y*	*Y*	*Y*	*Y*
*VIF*	<2	<2	<2	<2	<2	<2
*Adj-R²*	0.2216	0.0727	0.2511	0.1842	0.0891	0.3428
*F*	73.52 ***	20.97 ***	76.94 ***	91.73 ***	40.31 ***	187.29 ***
*Observations*	2039	2039	2039	3215	3215	3215
*Sobel*	0.00071 *** (*z* = 3.785)			−0.00027 (*z* = −1.429)		
*Goodman* 1	0.00071 *** (*z* = 3.766)			−0.00027 (*z* = −1.423)		
*Goodman 2*	0.00071 *** (*z* = 3.804)			−0.00027 (*z* = −1.435)		
Mediating effect	0.0007 *** (*z* = 3.785)			−0.00027 (*z* = −1.429)		
Direct effect	−0.0007 (*z* = −0.851)			0.00256 *** (*z* = 2.582)		
Total effect	−0.00001 (*z* = −0.020)			0.00229 ** (*z* = 2.274)		
Ratio of mediating effect	0.9757			0.1168		

Notes: Robust t-statistics are in parentheses; *, **, and *** represent significance at the levels of 10%, 5%, and 1%, respectively.

## Data Availability

Data is available upon request.

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
