# Peer review of "Mediation Effect of Corporate Tax Burden and the Relationship between Environmental Regulation and Firm Performance"

_ijerph, 2022, doi:10.3390/ijerph192214987_

Round 1

Reviewer 1 Report

1. On the abstract, the first sentence seems to suggest that you are using 2 dependent variables. May you kindly revisit this and align it with the rest of the document

3. A-share should be defined and well explained in the introduction

2. Line 27-28 and 110 to 112 should be referenced

3. In table 1, may you add a column showing the expected priori of the variables understudy

4. the empirical results were not compared to with the empirical literature. Kindly work on that. I see you have put some in the conclusion. I believe this should be under this section. The conclusion is for you to make conclusions

Author Response

Dear reviewer,

Thank you for your comments on our manuscript entitled “How Does Environmental Regulation Affect the performance of heavily Polluting Firms? Exploring the mediating effect of corporate tax burden” (ID: IJERPH- 2008895). Those comments are very helpful for revising and improving our paper. We have studied the comments carefully and made corrections which we hope meet with approval. The main corrections are in the manuscript and the responds to the reviewers’ comments are as follows (the replies are highlighted in red).

Replies to the reviewer’s comments:

Point 1: On the abstract, the first sentence seems to suggest that you are using 2 dependent variables. May you kindly revisit this and align it with the rest of the document

Response 1: We just used one dependent variable. We revised the first sentence on the abstract: This paper took the panel data of 1052 heavy-polluting listed companies from both Shanghai and Shenzhen Stock Exchange during 2010-2017 and employed mediation analysis method to empirically test the impact of environmental regulation on corporate performance and the mediating effects of corporate tax burden.

Point 2: A-share should be defined and well explained in the introduction

Response 2: A-share market is the Stock Exchange of Shanghai and Shenzhen in China, we have revised it in the text.

Point 3: Line 27-28 and 110 to 112 should be referenced

Response 3: We have added references in the line 27-28 and 110-112. The references as follows:

73.Ren, S., Wei, W., Sun, H., Xu, Q., Hu, Y., Chen, X., 2020. Can mandatory environmental information disclosure achieve a win-win for a firm’s environmental and economic performance? Journal of Cleaner Production, 250, 119530.

91.Porter, M.E., 1991. America’s green strategy. Scientific American 264, 168.

105.Zhou, D., Qiu, Y., Wang, M., 2021. Does environmental regulation promote firm profitability? Evidence from the implementation of China's newly revised Environmental Protection Law. Economic Modelling 102, 105585.

Point 4: In table 1, may you add a column showing the expected priori of the variables understudy

Response 4: We have revised table 1, added a column showing the expected priori of the variables understudy. The revised Table as follows:

Table 1. Definition of variables.

Variable

Category

Variable Names

Abbreviations

Measures

Expected priori

Dependent Variable

Firm performance

FP

Net profits divided by total assets

+

Independent Variable

Environmental regulation

ERG

Intensity index based on three types of waste emissions data

Mediating Variable

Corporate tax burden

ETR

Difference between tax expense and deferred tax provision divided by earnings before tax and interest

-

Control Variables

Operating income growth rate

GROWTH

The difference between the operating income of this year and the previous year divided by last year's operating income

+

Leverage

LEVE

Total liabilities divided by total assets at end of period

-

Cash flow

CASHF

Net cash flows from operating activities divided by operating income

+

R&D investment intensity

RD

Ratio of R&D investment to operating income

-

Book to market ratio

BM

Total assets divided by market value

-

Equity concentration

FIRST

Proportion of the largest shareholder’s stock holding

+

At the same time as chairman and general manager

DUAL

The chairman and general manager are assumed to be one by the same person, otherwise zero

-

Firm ownership

SOE

taking the value of one if the firm is a state-owned firm, zero otherwise.

+

Point 5: the empirical results were not compared to with the empirical literature. Kindly work on that. I see you have put some in the conclusion. I believe this should be under this section. The conclusion is for you to make conclusions

Response 5: We adjusted the structure of the paper, some of the discussion in the Conclusion section is moved to the Results section to facilitate the comparison. For details see the repaired paper.

Once again, thank you very much for your constructive comments and suggestions which would help us in depth to improve the quality of the paper.

Kind regards,

The authors

Reviewer 2 Report

REVIEWER’S COMMENTS TO THE AUTHOR

Recommendation: Minor Revision

The paper number ijerph-2008895 entitled “How Does Environmental Regulation Affect the performance of heavily Polluting Firms? Exploring the mediating effect of corporate tax burden” introduces an interesting topic.

The manuscript ijerph-2008895 is, however, lacking in some aspects. Here below, the authors can find some observations, suggestions and comments in order to improve the current version of the manuscript.

1. Originality:

Does the paper contain new and significant information adequate to justify publication?

Originality is not lacking in this manuscript.

2. Abstract:

The abstract effectively summarizes the contents of the paper, especially referring to its purposes, the methodology used, the originality of the research, and the key findings.

3. Introduction, Literature review and Methodology:

i) what literature says about this topic; ii) the existing gap in literature; iii) the aim of the research; iv) the methodology used; v) the main findings; vi) implications; vii) the remainder of the paper.

Except the below recommendations, these parts are well structured.

§ The authors are advised to update the literature and methodology sections by including related and recently published research (2020-2022) on the subject but not limited to the following:

https://doi.org/10.1007/s10668-020-01164-4

https://doi.org/10.1016/j.eiar.2022.106892

§ Also, the authors need to do an extensive review literature. The literature review should clearly establish the nexus between the independent variables and the dependent variable.

4. Results:

The results are presented clearly and analysed appropriately. The conclusions adequately tie together the other elements of the paper.

5. Implications for research, practice and/or society:

Does the paper identify clearly any implications for research, practice and/or society? Does the paper bridge the gap between theory and practice? How can the research be used in practice (economic and commercial impact), in teaching, to influence public policy, in research (contributing to the body of knowledge)? What is the impact upon society (influencing public attitudes, affecting quality of life)? Are these implications consistent with the findings and conclusions of the paper?

This part is well structured.

6. Quality of Communication:

Does the paper clearly express its case, measured against the technical language of the field and the expected knowledge of the journal’s readership? Has attention been paid to the clarity of expression and readability, such as sentence structure, jargon use, acronyms, etc.?

No, extensive proofreading is required.

Author Response

Dear reviewer,

Thank you for your comments on our manuscript entitled “How Does Environmental Regulation Affect the performance of heavily Polluting Firms? Exploring the mediating effect of corporate tax burden” (ID: IJERPH- 2008895). Those comments are very helpful for revising and improving our paper. We have studied the comments carefully and made corrections which we hope meet with approval. The main corrections are in the manuscript and the responds to the reviewers’ comments are as follows (the replies are highlighted in red).

Point 1: Introduction, Literature review and Methodology:

  1. i) what literature says about this topic; ii) the existing gap in literature; iii) the aim of the research; iv) the methodology used; v) the main findings; vi) implications; vii) the remainder of the paper.

Except the below recommendations, these parts are well structured.

  • The authors are advised to update the literature and methodology sections by including related and recently published research (2020-2022) on the subject but not limited to the following:

https://doi.org/10.1007/s10668-020-01164-4

https://doi.org/10.1016/j.eiar.2022.106892

  • Also, the authors need to do an extensive review literature. The literature review should clearly establish the nexus between the independent variables and the dependent variable.

Response 3: We have updated the literature and methodology sections by including more recently published researches (2020-2022) on the subject, including but not limited to 2 papers recommended by reviewers. Also, we add literature review, clearly establish the nexus between the independent variables and the dependent variable. For details see the repaired paper.

The references as follows:

3.Agyemang, A. O., Yusheng, K., Twum, A. K., Ayamba, E. C., Kongkuah, M., Musah, M., 2021. Trend and relationship between environmental accounting disclosure and environmental performance for mining companies listed in China. Environment, Development and Sustainability, 23(8), 12192-12216.

14.Chen, W., Chen, S., Wu, T., 2022. Research of the Impact of Heterogeneous Environmental Regulation on the Performance of China’s Manufacturing Enterprises. Frontiers in Environmental Science, 1091.

16.Deng, X., Huang, B., Zheng, Q., Ren, X., 2022. Can Environmental Governance and Corporate Performance be Balanced in the Context of Carbon Neutrality? —— A Quasi-Natural Experiment of Central Environmental Inspections. Frontiers in Energy Research, 10, 852286.

50.Long, F., Lin, F., Ge, C., 2022. Impact of China's environmental protection tax on corporate performance: Empirical data from heavily polluting industries. Environmental Impact Assessment Review, 97, 106892.

60.Mu, S., Wang, X., Mohiuddin, M., 2022. Impact of Environmental Protection Regulations on Corporate Performance From Porter Hypothesis Perspective: A Study Based on Publicly Listed Manufacturing Firms Data. Frontiers in Environmental Science, 10, 928697.

62.Park, W., Byun, C. G., 2021. Association between corporate tax burden and SMEs' future performance: Focus on KOSDAQ-listed firms. Academy of Strategic Management Journal, 20(5), 1-19.

78.Saragih, A. H., Hendrawan, A., Anggraini, P. G., Ayu, P. W. C., Dharsana, M. T., 2020. Can Earnings Performance Weaken the Negative Impact of Effective Tax Rate on Company’s Growth Prospects? An Empirical Study Based on Listed Firms of Financial Industry in Indonesia. Journal of Applied Economic Sciences, 3 (69):540-548.

81.Tao, A., Liang, Q., Kuai, P., Ding, T., 2021. The Influence of Urban Sprawl on Air Pollution and the Mediating Effect of Vehicle Ownership. Processes, 9(8), 1261.

86.Wang, Q., Xu, X., Liang, K., 2021. The impact of environmental regulation on firm performance: evidence from the Chinese cement industry. Journal of Environmental Management 299, 113596.

87.Wang, S., Cao, A., Wang, G., Xiao, Y., 2022. The Impact of energy poverty on the digital divide: The mediating effect of depression and Internet perception. Technology in Society, 68, 101884.

92.Xiang, D., Zhao, T., Zhang, N., 2022. How can government environmental policy affect the performance of SMEs: Chinese evidence. Journal of Cleaner Production, 336, 130308.

97.Yan, Y., Zhang, X., Zhang, J., Li, K., 2020. Emissions trading system (ETS) implementation and its collaborative governance effects on air pollution: The China story. Energy Policy, 138, 111282.

102.Zhang, Y., Wang, J., Chen, J., Liu, W., 2022. Does environmental regulation policy help improve business performance of manufacturing enterprises? evidence from China. Environment, Development and Sustainability, 1-30.

104.Zheng, H., He, Y., 2022. How do the China Pollution Discharge Fee Policy and the Environmental Protection Tax Law affect firm performance during the transitional period? Environmental Science and Pollution Research, 1-17.

Point 2: Quality of Communication:

Does the paper clearly express its case, measured against the technical language of the field and the expected knowledge of the journal’s readership? Has attention been paid to the clarity of expression and readability, such as sentence structure, jargon use, acronyms, etc.?

No, extensive proofreading is required.

Response 6: We have made extensive proofreading. For details see the repaired paper.

Once again, thank you very much for your constructive comments and suggestions which would help us in depth to improve the quality of the paper.

Kind regards,

The authors

Reviewer 3 Report

This paper has a potential to be accepted, but some important points have to be clarified or fixed before we can proceed and a positive action can be taken.

1-      The title is too long, it should be concise.

2-      Authors should revise better and more the current literature in the field.

3-      It would be wonderful if the shortcomings and gaps in the literature were clarified, particularly with regard to how the proposed strategy intends to fill up the gaps in the literature.

4-      The Tables need an explanation in the text! What is the message?

5-      Equations should be better defined including their necessity and relationship.

6-      It is advisable to offer a full experimental flowchart to do so because this will allow other researchers to more easily reference the suggested approach.

7-      There is a need to improve the analysis of outcomes based on the contributions of the manuscript.

Author Response

Dear reviewer,

Thank you for your comments on our manuscript entitled “How Does Environmental Regulation Affect the performance of heavily Polluting Firms? Exploring the mediating effect of corporate tax burden” (ID: IJERPH- 2008895). Those comments are very helpful for revising and improving our paper. We have studied the comments carefully and made corrections which we hope meet with approval. The main corrections are in the manuscript and the responds to the reviewers’ comments are as follows (the replies are highlighted in red).

Point 1:  The title is too long, it should be concise.

Response 1: We have revised the tittle: Mediation effect of corporate tax burden and the relationship between environmental regulation and firm performance.

Point 2: Authors should revise better and more the current literature in the field.

Response 2: We have revised added more current literature in the field.

The references as follows:

3.Agyemang, A. O., Yusheng, K., Twum, A. K., Ayamba, E. C., Kongkuah, M., Musah, M., 2021. Trend and relationship between environmental accounting disclosure and environmental performance for mining companies listed in China. Environment, Development and Sustainability, 23(8), 12192-12216.

14.Chen, W., Chen, S., Wu, T., 2022. Research of the Impact of Heterogeneous Environmental Regulation on the Performance of China’s Manufacturing Enterprises. Frontiers in Environmental Science, 1091.

16.Deng, X., Huang, B., Zheng, Q., Ren, X., 2022. Can Environmental Governance and Corporate Performance be Balanced in the Context of Carbon Neutrality? —— A Quasi-Natural Experiment of Central Environmental Inspections. Frontiers in Energy Research, 10, 852286.

50.Long, F., Lin, F., Ge, C., 2022. Impact of China's environmental protection tax on corporate performance: Empirical data from heavily polluting industries. Environmental Impact Assessment Review, 97, 106892.

60.Mu, S., Wang, X., Mohiuddin, M., 2022. Impact of Environmental Protection Regulations on Corporate Performance From Porter Hypothesis Perspective: A Study Based on Publicly Listed Manufacturing Firms Data. Frontiers in Environmental Science, 10, 928697.

62.Park, W., Byun, C. G., 2021. Association between corporate tax burden and SMEs' future performance: Focus on KOSDAQ-listed firms. Academy of Strategic Management Journal, 20(5), 1-19.

78.Saragih, A. H., Hendrawan, A., Anggraini, P. G., Ayu, P. W. C., Dharsana, M. T., 2020. Can Earnings Performance Weaken the Negative Impact of Effective Tax Rate on Company’s Growth Prospects? An Empirical Study Based on Listed Firms of Financial Industry in Indonesia. Journal of Applied Economic Sciences, 3 (69):540-548.

86.Wang, Q., Xu, X., Liang, K., 2021. The impact of environmental regulation on firm performance: evidence from the Chinese cement industry. Journal of Environmental Management 299, 113596.

92.Xiang, D., Zhao, T., Zhang, N., 2022. How can government environmental policy affect the performance of SMEs: Chinese evidence. Journal of Cleaner Production, 336, 130308.

102.Zhang, Y., Wang, J., Chen, J., Liu, W., 2022. Does environmental regulation policy help improve business performance of manufacturing enterprises? evidence from China. Environment, Development and Sustainability, 1-30.

104.Zheng, H., He, Y., 2022. How do the China Pollution Discharge Fee Policy and the Environmental Protection Tax Law affect firm performance during the transitional period? Environmental Science and Pollution Research, 1-17.

Point 3:  It would be wonderful if the shortcomings and gaps in the literature were clarified, particularly with regard to how the proposed strategy intends to fill up the gaps in the literature.

Response 3: We have clarified the shortcomings and gaps in the literature at page 2, line 74-79. Details as follows: “As an important factor affecting firm performance, corporate tax is one of the widely used instruments for the government to implement environmental regulation, which has strong incentives for firm's behavior. However, there are limited literature aiming at analyzing the impact of environmental regulation via efficient tax rate. To fill this gap, in this paper we explore the impacts of environmental regulation on firm performance, assess the mediating effects of efficient tax rate and try to provide more empirical evidence for the reference of policymakers.”

Point 4:   The Tables need an explanation in the text! What is the message?

Response 4: We have made explanations in the text about Tables. For details see the repaired paper.

Point 5:  Equations should be better defined including their necessity and relationship.

Response 5: We have better defined the Equations including their necessity and relationship. For details see the repaired paper.

Point 6:  It is advisable to offer a full experimental flowchart to do so because this will allow other researchers to more easily reference the suggested approach.

Response 6: We have added a full technical flowchart. The revised Figure as follows:

Figure 2. Technical flowchart.

Point 7: There is a need to improve the analysis of outcomes based on the contributions of the manuscript.

Response 7: We have improved the analysis of outcomes and add some discussions that links the empirical findings with the contribution of the study. The changes can be found at page 11-12, line 451-458, page 13, line 480-481 and page 14, line 542-543.

Once again, thank you very much for your constructive comments and suggestions which would help us in depth to improve the quality of the paper.

Kind regards,

The authors
